# Multiplex Digital PCR to Detect Amplifications of Specific Androgen Receptor Loci in Cell-Free DNA for Prognosis of Metastatic Castration-Resistant Prostate Cancer

**DOI:** 10.3390/cancers12082139

**Published:** 2020-08-01

**Authors:** Meijun Du, Chiang-Ching Huang, Winston Tan, Manish Kohli, Liang Wang

**Affiliations:** 1Department of the Genomic Sciences and Precision Medicine Center (GSPMC), Medical College of Wisconsin, Milwaukee, WI 53226, USA; dumeijun1020@gmail.com; 2Department of Pathology, MCW Cancer Center, Medical College of Wisconsin, Milwaukee, WI 53226, USA; 3Department of Biostatistics, Zilber School of Public Health, University of Wisconsin, Milwaukee, WI 53205, USA; huangcc@uwm.edu; 4Department of Medicine, Division of Hematology/Oncology, Mayo Clinic, Jacksonville, FL 32224, USA; tan.winston@mayo.edu; 5Division of Oncology, Department of Medicine, University of Utah, Huntsman Cancer Institute, Salt Lake City, UT 84112, USA; 6Department of Tumor Biology, H. Lee Moffitt Cancer Center, Tampa, FL 33612, USA

**Keywords:** metastatic castration-resistant prostate cancer, multiplex digital PCR, copy number, androgen receptor, cell-free DNA, overall survival

## Abstract

Amplification of androgen receptor (*AR*) is a common genomic event in metastatic castration-resistant prostate cancer (mCRPC). To evaluate the prognostic value of the amplifications of specific loci in the *AR* gene in cell-free DNA, we developed a multiplex digital PCR (dPCR) assay that targeted *AR* enhancer (*AR*-En), *AR* exon 1 (*AR*-E1), *AR* exon 8 (*AR*-E8) and *OPHN1* (downstream of *AR*). We selected three relatively stable genes, *C2orf16*, *FAM111B*, and *GRIA3,* as reference controls for copy number normalization. One hundred and eight mCRPC patients were recruited to test the association of specific AR loci amplification with clinical outcome. Using a normalized ratio ≥ 1.92 as cutoff, amplification of *AR*-En, *AR-*E1, *AR*-E8 and *OPHN1* was observed in 28, 25, 24 and 19 of 108 mCRPC patients, respectively. Among the 41 patients with *AR* region amplification, 9 (21.9%) showed amplification at all four selected regions and 15 (36.6%) showed amplification at *AR*-En, *AR-E1,* and *AR-*E8. Six (14.6%) patients showed independent *AR*-En amplification, while the remaining 3 (7.3%) demonstrated *AR*-E8 amplification only. Kaplan–Meier analysis showed overall survival’s association with the amplification of *AR*-En (*p* = 0.02, HR = 1.68 (1.07–2.65)), *AR*-E8 (*p* = 0.02, HR = 1.78 (1.08–2.92)) and *AR*-En-E8 (the combination of *AR*-En and *AR*-E8 (*p* = 0.009, HR = 1.77 (1.15–2.73)). Multivariate models that included *AR*-En-E8 amplification and clinical factors significantly improved prognostic performance (*p* = 0.0001). With further validation, the multiplex dPCR assay may assist in prognostication of mCRPC patients.

## 1. Introduction

Circulating cell-free DNA (cfDNA) has become a promising tool in molecular oncology, allowing the detection of molecular alterations associated with cancer biology, treatment response [1,2] and overall survival (OS) [3,4,5,6]. However, cfDNA analysis is subject to relevant limitations [7]. There still remain technical challenges to analyze small amounts of highly fragmented (150–200 bp) and diluted (nanograms per 1 mL plasma) cfDNA fractions that are attributable to tumors (circulating tumor DNA-ctDNA) within cfDNA that are shed from cancer cells and which generally comprise 1–2% or less of cfDNA. In addition, the sensitivity of sequencing techniques is limited by the availability of an adequate amount of ctDNA. To increase the sensitivity of detecting ctDNA in the blood stream, digital PCR (dPCR) has been developed, which allows the detection of mutant and wild type DNA fragments at ratios close to 1:100,000 (allelic frequency (AF) = 0.001%) [8,9,10]. In addition, by varying parameters that affect PCR efficiency and end-point fluorescence, dPCR has been used to detect multiple targets in one reaction with only two different fluorophores [11,12].

Prostate cancer is one of the leading causes of cancer death in males in the Western world [13], which occurs typically in the advanced, metastatic castration-resistant prostate cancer (mCRPC) state. The mCRPC state is highly heterogeneous in its clinical behavior, ranging from slowly progressing disease with long-lasting responses to anti-androgen therapy to rapidly progressive lethal tumors characterized by early treatment resistance to available therapies. With the increasing number of options for the treatment of mCRPC [14,15], it is urgent to identify molecular and genetic biomarkers to optimize the treatment and guide personalized therapy. A well-known genomic alteration in mCRPC is that the androgen receptor (*AR)* gene is amplified in nearly half of mCRPC cases [16,17,18,19,20]. Increased *AR* activity can drive therapeutic resistance in advanced prostate cancer [21,22], rendering several novel therapeutics ineffective. Recently, it was reported that a somatically acquired *AR* enhancer region, 650 kb upstream of *AR*, drives the progression of mCRPC [18,23,24]. We recently reported that *OPHN1* gene, which is downstream of *AR,* is also amplified in mCRPC [25].

In this study, we systematically determined the frequencies and association of amplifications in multiple loci in and near the *AR* gene including *AR*-En and *OPHN1* to determine the impact on survival. For this, two multiplex dPCR assays were developed that targeted four selected loci including two targets genes and two normalization controls in each assay by adjusting concentrations of primers and probes and the type of fluorophores. We then applied the assays to analyze the amplification status of *AR*-En, *AR* exon 1 (*AR*-E1), *AR* exon 8 (*AR*-E8), and *OPHN1* in mCRPC patients receiving chemotherapy as a clinically useful molecular biomarker.

## 2. Results

### 2.1. Clinical Characteristics of Patient Samples

One hundred and eight mCRPC patients were enrolled between September 2009 and July 2013 and followed until the date of cut off for analysis (1 October 2018). After failure of androgen-deprivation therapy (ADT), 59/108 patients received systemic chemotherapy (majority being with docetaxel). One hundred and two died during follow-up with a median follow-up of 92.2 (range, 64.6–109.5) months. At the time of study enrollment to the mCRPC cohort, the median prostate specific antigen (PSA) and serum alkaline phosphatase (ALP) levels were 16.7 ng/mL (range, < 0.1–2324) and 94 IU/L (range, 39–2185), respectively. The median time for ADT failure prior to progression to mCRPC state of the study cohort was 19.7 months (range, 0.7–202.2). Clinical characteristics of the study cohort are listed in Table 1.

### 2.2. Stability of Multiplex dPCR Assays in Normal Controls

We first established the multiplex dPCR assay in prostate cancer cell line DNA (LNCaP and 22RV21) using duplex qPCR by combining single fluorophore FAM and HEX-labeled probes. To separate two targets with the same fluorophore, we adjusted the probe ratio, with one using normal dilution (1×) and another using 40% of normal dilution (0.4×). To balance the signal density of different fluorescence, we applied 1:1.5 ratio between FAM and HEX-labeled probes. Each sample was analyzed by two assays. Assay I included target genes *AR*-E1, *AR*-E8, and control *FAM111B*, *GRIA3*; Assay II included *OPHN1*, *AR*-En and control *FAM111B*, *C2orf16.* Both assays shared a common control to normalize potential sample loading bias (Figure 1A,B). To test the stability and consistency of the established assay, we performed the multiplex dPCR experiments using 14 normal genomic DNA (gDNA) as input and calculated the normalized copy number (CN) for each of the targets, including *AR*-En, *AR*-E1, *AR*-E8, and *OPHN1*. After normalizing to the average of three reference controls, the mean CN ratio for each locus was: 1.01 (range 0.84–1.32) for *AR*-En, 0.92 (Range 0.69–1.30) for *AR*-E1, 1.08 (Range 0.86–1.41) for *AR*-E8, and 1.12 (range 0.96–1.29) for *OPHN1*, indicating the consistency of CN detection for these targets (Appendix A).

### 2.3. Multi-Loci Reference Controls

To select best reference controls for CN quantification at *AR* loci, we first examined existing databases [16,26,27,28] to determine the relative stable genomic regions in prostate cancer. This analysis showed the most stable regions in three gene loci, including *C2orf16* at 2p23.2, *FAM111B* at 11q12.1, and *GRIA3* at Xq25. We tested the stability of these three genes in the 108 plasma cfDNA dPCR data using software RefFinder and found their stability in order of *FAM111B* > *GRIA3 > C2orf16.* However, the most stable reference control was found by the combination of *FAM111B*, *GRIA3*, and *C2orf16* (Appendix A). We further analyzed the correlation of the CN between different reference controls in the same assay and between the two different assays. We observed R^2^ = 0.80 for *FAM111B* and *GRIA3* in Assay I, R^2^ = 0.80 for *FAM111B* and C2orf26 in Assay II, and R^2^ = 0.93 for *FAM111B*-Assay I and *FAM111B*-Assay II (Appendix A). Considering the stability and heterogeneity, we used the average of *FAM111B*, *C2orf16*, and *GRIA3* to normalize the CNs of *AR*-En, *AR-*E1, *AR*-E8, and *OPHN1* for all plasma samples.

### 2.4. Amplifications at AR Loci in Plasma cfDNA

To evaluate the CN ratio at *AR*-En, *AR-*E1, *AR*-E8, and *OPHN1,* we applied the optimized multiplex dPCR assays to test the cfDNAs from 108 mCRPC patients. Figure 1C,D show an example of amplification of *AR*-En, *AR*-E1, and *AR*-E8 but not *OPHN1*. After normalization to the averaged combined reference control, we observed the median Log2 CN ratio, being 0.52 (range −1.42–6.00) for *AR*-En, 0.29 (range −1.69–5.45) for *AR*-E1, 0.36 (range −0.92–5.45) for *AR*-E8, and 0.14 (range −1.54–5.04) for *OPHN1* (Figure 2A). We defined the amplification as the normalized CN ratio ≥ 1.92 between the targets and reference loci using a suggested threshold [29] and our previously reported observations made in an independent mCRPC cohort [4]. Among the 108 mCRPC patients, 28 (25.9%) had *AR* enhancer amplification; 24 (22.2%) had *AR-*E8 amplification; 25 (23.1%) had *AR*-E1 amplification; and 19 (17.6%) had *OPHN1* amplification (Figure 2B). Importantly, among the 41 of 108 (38.9%) *AR* loci amplified patients (at least one of the targets amplified), we observed that 9 patients shared *AR*-En, *AR-*E1*, AR-*E8 and *OPHN1* amplifications; 15 patients shared *AR*-En, *AR-*E1*,* and *AR-*E8 amplification; 6 patients had independent *AR*-En amplification; 3 patients had independent *AR*-E8 amplification; and 2 patients had independent *OPHN1* amplification (Figure 2C). We further checked the consistence of the AR-dPCR using low-pass whole genome sequencing data in 13 samples with co-amplification of *AR-En*, *AR*-E1, and *AR*-E8. The heatmap (1 Mb genomic window covering *AR-En*, *AR*-E1, and *AR*-E8) showed clear amplification of AR region in 12 of the 13 samples (Appendix A), supporting the observations made for AR amplification by the dPCR assay.

### 2.5. Association of AR Loci Amplification with Clinical Outcomes

We checked the association of *AR* loci amplification with clinical indices of PSA, ALP, time to ADT failure, and Gleason score using Fisher’s exact test. Patients were divided into two groups based on Gleason score > 8, above or less than the median PSA at the time if mCRPC state enrollment (>16.7 ng/mL), less than or greater than the median time to ADT failure (<19.7 months), and above or below the median ALP (>94 IU). We found statistically significant correlation between plasma *AR* amplifications and clinical indices of time to ADT failure (*p* = 0.0052), and Gleason score (*p* = 0.0086). No significant association between *AR* amplification and PSA and ALP level (*p* > 0.1) was found. To test the association of the CN with overall survival (OS), we performed K–M analysis using the defined amplification of *AR-En*, *AR-*E1*, AR-*E8 and *OPHN1* gene loci. Significant association of amplification in *AR*-En (*p* = 0.02, HR = 1.68 (1.07–2.65)), *AR*-E8 (*p* = 0.02, HR = 1.78 (1.08–2.92)) with OS were observed. Patients harboring *AR*-*En* and *AR*-E8 amplification had a significantly shorter OS. The median OS was 19 vs. 27 months for *AR*-En amplification-positive and -negative patients and 21 vs. 27 months for *AR*-E8 amplification-positive and -negative patients, respectively (Figure 3A,B). No significant association of *AR*-E1 (*p* = 0.08) and *OPHN1* (*p* = 0.23) (Appendix A) was observed using the cut off ratio of 1.92 for *AR* CN amplification. However, when AR CN ratio for amplification call was reduced from 1.92 to 1.8 for *AR*-E1, we observed a significant association at the target genomic region with OS (Appendix A). Considering the genetic heterogeneity of *AR* loci, we further combined patients with either *AR*-En or *AR*-E8 amplifications and performed association analysis with OS. A significant association with OS was observed when *AR*-En and *AR*-E8 amplified samples were combined (*AR*-En-E8) (*p* = 0.009, HR = 1.77 (1.15–2.73)) (Figure 3C).

### 2.6. Multivariate Cox-Regression Model Predict OS

To evaluate the added prognostic value of these molecular markers, we constructed a multivariate Cox model that included additional clinical parameters. We first evaluated the association of clinical factors (PSA levels at the time of enrollment, ALP, time to ADT failure, Gleason Score, age, and bone metastasis) with OS. At the univariate Cox-regression level, shorter time to ADT-failure (*p* = 0.004), and greater than median ALP levels (*p* = 0.005) were statistically associated with poor OS, but not Gleason score, age and presence of bone metastasis (*p* > 0.1). We then constructed a multivariate model, which included molecular (CN of *AR*-En-E8) and clinical prognostic factors significantly associated with survival at the univariate levels. The results indicate that *AR*-En-E8 amplification, shorter time to ADT failure, and higher ALP were all independently significantly associated with poor OS (Table 2). We created a risk index as a weighted score of *AR*-En-E8 amplification, time to ADT failure (short vs. long), and level of ALP, where the weights were the regression coefficients from the multivariate Cox model. We found that the risk group based on a median risk index cut point was significantly associated with OS (*p* = 0.0001, HR = 1.90 (1.25 to 2.88)) (Figure 3D).

## 3. Discussion

*AR* amplification has been previously reported in several independent studies to prognosticate survival in mCRPC state, and our group recently also reported AR-En amplification was associated with primary resistance to abiraterone acetate treatments [4,25,30]. We now report a quantitative, sensitive four CN altered targets-based multiplex dPCR assay to detect CN changes in *AR*-En, *AR-*E1, *AR*-E8, and *OPHN1* genes using cfDNA from plasma of mCRPC patients. By varying the concentrations of primers and probes and the type of fluorophores, accurate, precise, and absolute quantification of the specific nucleic acid sequences and the high sensitivity of dPCR, we were able to detect low frequency of tumor DNA based on alterations among the excessive background of wild-type DNA in cfDNA samples. We also observed that the increased number of potential targets per test significantly improved dPCR clinical utility, by reducing the cost and also improving the CN alteration output information from a single plasma specimen. These results suggest that a multiplex dPCR assay can be used as a potentially valuable molecular biomarker assay to detect genomic alterations in mCRPC patients.

We have previously reported amplification frequencies in plasma cfDNA for *AR-* and *OPHN1* in a different cohort of mCRPC patients by using low-pass whole genome sequencing (WGS) [25]. In the current study, we tested the amplification frequencies in plasma cfDNA for *AR*-En, *AR-*E1, *AR-*E8, and *OPHN1* by multiplex dPCR. We observed that the amplification frequencies of *AR*-En was more frequent than *AR* itself and *OPHN1*. While most of the cases with *AR* enhancer, *AR-E1*, *AR*-E8 and/or *OPHN1* were co-amplified, six patients with *AR*-En and three with *AR*-E8 were independently amplified, possibly indicating an independent interplay of metastasis driver for both *AR* and *AR* enhancer, which is consistent with previous reports [18,23,24]. In comparison to dPCR, amplification calls made using WGS often uses genomic bins (from 100 kb to 1 mb in size, depending on the read depth) which covers target genes as well as multiple other genes. The lack of specificity of target regions using a bin-based method may dilute the amplification signal and in turn impact clinical utility. Therefore, the low pass WGS determinations not only tend to have lower resolution, but also reduced sensitivity. Our study suggests that the dPCR assay may narrow down the amplification region to the specific gene region (even domain region such as enhancer) by designing primers for the exact genomic locations of the selected targets.

Previous studies have shown that 30–50% mCRPC patients have *AR* amplification [16,31], and in one recent study in 40 mCRPC patients *AR* and *AR*-En amplifications were determined in cfDNA by using targeted sequencing [32]. The rate of *AR* and *AR*-En amplifications was detected to be 45% and 40%, respectively, and was correlated with poor survival. Using a dPCR-based assay, we observed lower amplification frequency (about 20–25%) in our cohort. There may be several factors for this difference which could contribute to the low detectable amplification in our study. One key factor is the cutoff call to define *AR* amplification. Because the cutoff value (normalized ratio) determines amplification status, the number of patients with *AR* amplification can be significantly increased if lower cutoff value is applied. In fact, if, in our study, the cutoff value is set to 1.8, the number of patients with amplification at *AR* loci is observed to increase to 46 (42.5%), which is similar to other reports [16,31]. We observed that with this lower cutoff value for amplification calls, the survival association remained significant. Interestingly, the association of AR-E1 with OS was changed to significant from insignificant (*p* value from 0.08 was changed to 0.02). However, to stay consistent with the recently published cutoff call rates on amplification status [29], we selected 1.92 as the cutoff value for amplification calls.

The reference controls selected in our study were based on the gene loci that are relatively stable in prostate cancer based on published databases. Our results show that the average of the three controls was more stable than any individual one. By combining the three reference loci, the assay can further minimize bias caused by potential CN changes at these regions. Additionally, the information derived from combination of target loci (*AR*-En and *AR*-E8) potentially improves the sensitivity of the test and also informs about inherent tumor heterogeneity in the mCRPC patient population. We expect that applying multiplex dPCR to other genomic loci, such as *ZFHX3* and *PIK3CA,* in the mCRPC stage will show significant associations with treatment response [25], and may extend clinical value for patient management.

Although promising, this plasma cfDNA-based multiplex dPCR analysis has its limitations. First, we used the pre-amplified DNA as a PCR template, which may generate CN detection bias. For clinical application, the original cfDNA directly from plasma should be used. Second, although adjusting the concentrations of primers and probes and the type of fluorophores could separate the different signal clusters in the multiplex dPCR assays in most samples, some signal overlaps in a few samples were also observable, which may cause bias for CN detection. Third, we focused on four targets at *AR* locus only and while some patients demonstrated an independent amplification of the *AR*-En and *AR*-E8 loci, most of the patients showed co-amplification. This may be due to the inherent heterogeneity of prostate cancer biology and so the use of combinations of the *AR* loci across genomic regions over a single gene level readout may increase the predictive performance of this assay as a molecular biomarker.

## 4. Patients and Methods

### 4.1. Patient Samples

Plasma specimens were obtained from 108 mCRPC patients, enrolled in a Mayo Clinic Institutional Board (09-001889)-approved hospital-based cohort study from September 2009 to July 2013, following androgen-deprivation therapy (ADT) failure, as was previously described [33,34]. All patients provided written informed consent. All patients in the study were followed until death or censored at last follow-up, the date for which was 31 October 2018. Details of this hospital-based prospectively collected repository were previously published [33,34].

### 4.2. Multiplex dPCR

cfDNA was extracted from plasma and pre-amplified by using a ThruPlex DNA-Seq Kit (Rubicon Genomics, Ann Arbor, MI, USA) as previously described [30]. The target regions were selected based on our previous sequencing data showing amplification at genomic *AR* loci, including *AR*-En, *AR*, and *OPHN1* in mCRPC patients [25]. Three genes—*C2orf16* at chr2p23.3, *FAM111B* at chr11q12.1, and *GRIA3* at chrXq25—were selected as controls for CN normalization based on the CN stability of these genes in four different prostate cancer databases [16,26,27,28]. The primers and probes were designed using Primer Premier 5 (San Francisco, CA, USA) and synthesized in Integrated DNA Technologies (IDT, Coralville, IA, USA). TaqMan probes were labeled with 5′ FAM (targets) or HEX (controls). The sequences of the primers and probes are listed in Appendix A. Then, 20 × primers/probes master mix was prepared with 5 μM of primer pairs (*AR*-En, *AR*-E1, *AR*-E8, *OPHN1*, *C2orf26*, *FAM111B*, *GRIA3*) and 3μM of probes.

We performed two dPCR assays for each individual sample. Each assay included two target genes, *AR-*E1 and *AR-*E8 in Assay I, and *AR*-En and *OPHN1* in Assay II, respectively. Two CN control primer/probe pools were added for CN determination, which were *FAM111B* and *GRIA3* in Assay I and *FAM111B* and *C2orf26* in Assay II, respectively. QuantStudio 3D Digital PCR System (Life Technology) was used for the dPCR. For each chip, reactions were performed in 15.5 μL volume using 7.25 μL of 2 × 3D Digital PCR master mix, 1.5 μL High Hex 20 × primers/probes master mix, 0.6 μL low Hex 20 × primers/probes master mix, 1 μL high FAM 20 × primers/probes master mix and 0.4 μL low FAM 20 × primers/probes master mix, respectively. Reactions were performed under universal cycling conditions: 96 °C for 10 min, followed by 45 cycles at 58/60 °C for 2 min and 98 °C for 30 s with a final extension at 60 °C for 2 min.

### 4.3. Quantification of CN of Different Targets at AR Loci

The chip signal image was captured by the QuantStudio 3D Digital PCR system. Data analysis was performed using the AnalysisSuite Software (Life technology). The number of target molecules was calculated using the Poisson distribution, which provided the CN per μL reaction mix. The CN ratio of each target at AR loci was estimated by normalizing to the average of three reference genes. The formula is: CN on chrX = target copies/μL*2 divided by control copies/μL.

### 4.4. Confirmation of AR Amplification by Low-Pass Whole Genome Sequencing

Thirteen of 108 samples were selected for low-pass whole genome sequencing to confirm AR amplification. cfDNA libraries (ThruPlex DNA-Seq Kit) were sequenced in Illumina HiSeq2500 for single-end 50 bp read. Sequencing reads from FastQ files were first mapped to human genome (hg19) and then summarized into 1 Mb genomic bins, which were further normalized to a group of 15 healthy individuals by log2 ratio transformation. The log2 ratios with 1Mb bin size at AR and its flanking regions were analyzed for CN changes. Detail methods for CN analysis have been published previously [25].

### 4.5. Statistical Analysis

The primary aim of the study was to identify associations between plasma *AR* loci CN aberrations and overall survival (OS). OS was calculated from the time from the date of study enrollment at the time of progressing to mCRPC state to the date of death or the date of the last follow-up (30 October 2018). Association of CNs with clinical factors were tested with Fisher’s exact test. Association between CNs and OS was assessed using Kaplan–Meier (K-M) survival curves. Tests of significance for amplification and association with OS were performed using log-rank test with statistical significance set at *p* ≤ 0.05. To determine the effect of multiple factors on survival, a multivariate Cox regression model was utilized to assess association of several covariates measured at study enrollment with OS. These included *AR* loci amplification, PSA, serum ALP, and time to ADT failure. To avoid over-parameterization during the multivariate modeling process due to a relatively small number of patients and large number of potential covariates, the final multivariate model only fitted factors with an entry threshold of *p* ≤ 0.05 in univariate analysis.

## 5. Conclusions

We reported a multiplex dPCR assay to detect CN changes at *AR* loci in plasma cfDNA and highlight its potential clinical use as non-invasive molecular testing tool for mCRPC prognosis. We were able to successfully use three relatively stable genes as reference controls to minimize potential error by single reference gene assays and were able to probe a combination of *AR* target regions to increase the detection sensitivity of these *AR* loci. We anticipate that further development of the easy-to-use and low-cost liquid biopsy assay will facilitate its clinical application in the highly heterogeneous mCRPC patients.

## Figures and Tables

**Figure 1 cancers-12-02139-f001:**
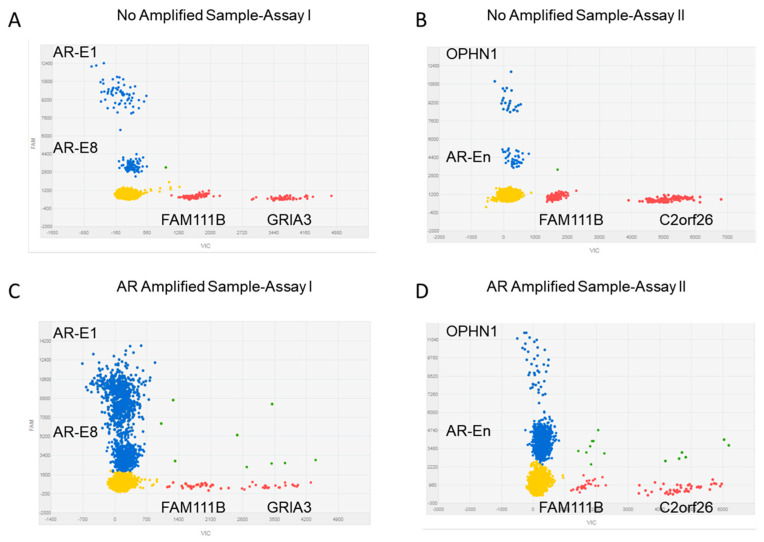
Simultaneous detection of two targets and two controls by multiplex dPCR assay using fluorophore combinations and ratio dilution of the same fluorophore. The figures given here used the QuantStudio 3D Digital PCR system (Life Technology, Carlsbad, CA, USA). For each fluorophore FAM and HEX combination and ratio-based multiplexing dPCR assay, the configuration of clusters in the 2D plot is given with the two clusters in FAM (blue) on the y-axis and two clusters in HEX (Red) on the x-axis. (**A**) FAM signals represent two independent targets: *AR*-E1 (high), *AR*-E8 (low), HEX signals represent two independent controls *FAM111B* (low) and *GRIA3* (high). (**B**) FAM signals represent another two independent targets: *OPHN1* (high), *AR*-En (low) and HEX signals represent another two independent controls: *FAM111B* (low) and *C2orf26* (high). The x-axis displays the amplitude of genome CN controls (labeled by HEX, red) and the y-axis represents the signals of targets genome CN (labeled by FAM, blue). The signals in the left quadrant are negative for both targets (yellow). The signals in the upper right quadrant are positive for both targets (green). (**C**,**D**) represent one amplified sample at *AR* loci for amplification of *AR*-E1, *AR*-E8 and *AR*-En, but not *OPHN1.*

**Figure 2 cancers-12-02139-f002:**
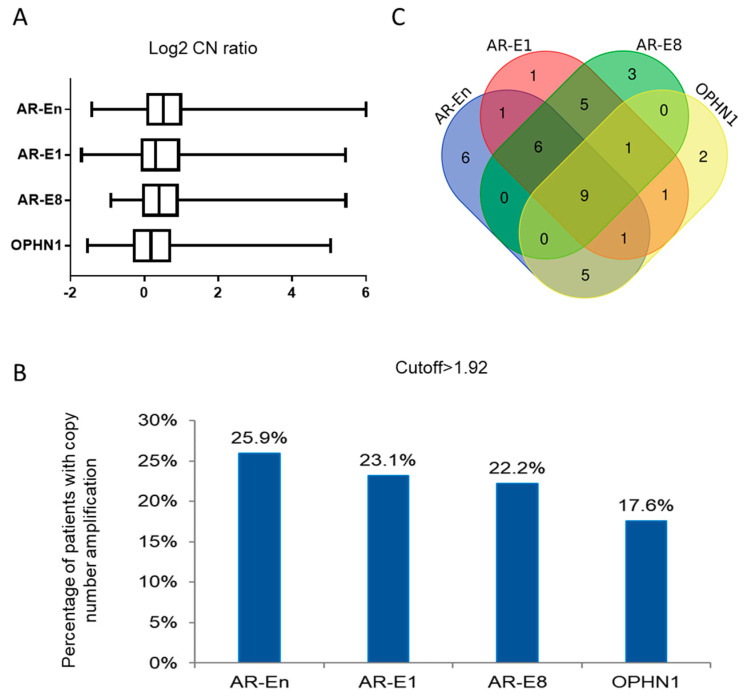
Distribution of amplifications status at *AR*-En, *AR-*E1*, AR-*E8, and *OPHN1*. (**A**) Box and whiskers plot shows the range of log2 ratio of plasma CN of *AR*-En, *AR*-E1, *AR*-E8, and *OPHN1* in 108 mCRPC patients as determined by CNs of targets divided by CNs of references genes. (**B**) Percentage of amplifications at *AR*-En (28), *AR*-E1 (25), *AR*-E8 (24), and *OPHN1* (19) in 108 mCRPC patients. (**C**) Venn dendrogram shows the distribution, co-amplification, and independent amplification of *AR*-En, *AR*-E1, *AR*-E8, and *OPHN1* in 41 *AR* loci amplified patients.

**Figure 3 cancers-12-02139-f003:**
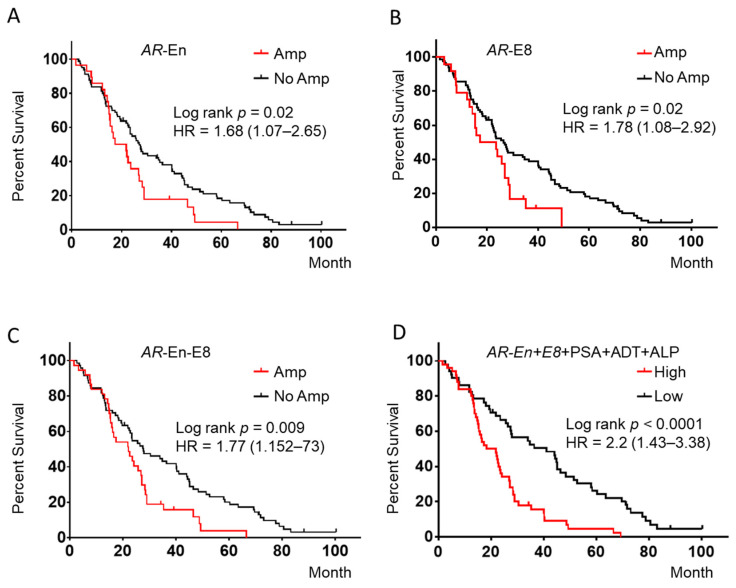
Association of the amplification at *AR*-En, *AR-*E8, *AR*-En-E8, and the multivariate model with OS. (**A**) *AR*-En amplification was associated with overall survival (OS). (**B**) *AR*-E8 amplification was associated with OS. (**C**) The combination of *AR*-En and *AR*-E8 was significantly associated with OS. (**D**) Multivariate model (*AR*-En-E8 + ADT + ALP) showed significant association with OS. ADT: time to ADT failure.

**Table 1 cancers-12-02139-t001:** Clinical characteristics of the metastatic castration-resistant prostate cancer (mCRPC) cohort.

Clinical Variables	Total (*n* = 108)
Age-year	
Median (Range)	75.4 (44.6–93.2)
Patients who received radiation therapy after initial diagnosis, no. (%)	25 (23.1)
Patients with previous history of radical prostatectomy after initial cancer diagnosis, no. (%)	55 (50.9)
Patients who received no primary prostate cancer treatments as initial diagnosis was with metastases, no. (%)	35 (32.4)
Data missing on primary prostate cancer therapy	8
Gleason score at initial diagnosis (%); Pathological	
(Pathological Gleason scores for the 54 patients with Radical Prostatectomy)	
Gleason score at initial diagnosis-no. (%)	
5–6	12 (11.1)
7	39 (36.1)
8–10	49 (45.4)
Missing	8 (7.4)
Mean * Basal Metabolic Index (BMI) at the time of enrollment	
Median, Range (*n* = 2 missing)	28.9 (20.1–44.8)
Patients with previous Radical Prostatectomy no. (%)	55 (50.9)
Patients with previous Radiation therapy no. (%)	26 (24.1)
Time from initial prostate cancer therapy to start of ADT (Months)	
Median (range)	45.8 (0.07–222.7)
Time from initial prostate cancer therapy to progression to mCRPC state, (Months)	
Median (range)	89.9 (7.5–281.0)
Patients (out of 108) who underwent salvage/adjuvant therapies for progression after primary prostate therapy no. (%)	60 (55.6)
Patients who underwent secondary hormonal maneuvers after failure of primary ADT no. (%)	86 (79.6)
Patients who underwent systemic chemotherapy for CRPC state no. (%)	59 (54.6)
PSA at date of mCRPC enrollment, ng/mL	
Median(range)	16.7(<0.1–2324)
ALP levels at date of mCRPC enrollment, IU/L	
Median(range)	94(39–2185)
Time from initiation of ADT to progression for CRPC stage, (Month)	
Median(range)	19.7(0.7–202.2)
Bone metastasis, no. (%)	
Yes	95 (88.0)
No	12 (11.1)
Missing	1 (0.9)
Follow-up time from date of CRPC specimen collection to last follow-up or death (Months)	
Median(range)	92.2 (64.6–109.5)
Number of deaths during follow up	102

Abbreviations: mCRPC, metastatic castration-resistant prostate cancer; ADT, androgen-deprivation therapy; PSA, prostate-specific antigen; ALP, alkaline phosphatase.

**Table 2 cancers-12-02139-t002:** Multivariable Cox proportional hazards analysis of predictors for OS.

Variables	HR	95% CI		*p*-Value
AR-En-E8-Amplification present	1.676	1.038	2.707	0.035
Log_Time to ADT Failure	0.786	0.641	0.963	0.020
Log_ALP	1.553	1.152	2.094	0.004

Abbreviations: *AR*-En-E8, androgen receptor enhancer and/or exon8 amplified samples; ADT, androgen-deprivation therapy; ALP, alkaline phosphatase.

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
