# Peer review of "Multiplex Digital PCR to Detect Amplifications of Specific Androgen Receptor Loci in Cell-Free DNA for Prognosis of Metastatic Castration-Resistant Prostate Cancer"

_cancers, 2020, doi:10.3390/cancers12082139_

Round 1

Reviewer 1 Report

The authors analyzed amplification of androgen receptor (AR) and OPHN1 genes from circulating cell-free DNA in metastatic castration-resistant prostate cancer (mCRPC), and detected that amplification of some locations of AR gene was associated with overall survival. Their results show that prognostic analysis can be performed using the serum without the use of biopsy or other invasive sample collection. Therefore, it may become a good method for the prognosis for prostate cancer patients. There are some questions and suggestions as described below.

Major comment:

  • Authors checked 3 locations of AR gene among the mCRPC patients in this study. The amplification of locations of AR enhancer (AR-En) and AR exon 8 (AR-E8) was associated with overall survival, but not that of AR exon 1 (AR-E1). Because AR-E1 locates between AR-En and AR-E8, it is difficult to understand why the amplification of location of AR-E1 was not associated with overall survival. Interestingly, when both locations of AR-En and AR-E8 were amplified, the location of AR-E1 was also amplified in all 15 patients of this study. How did authors think about these phenomena? They should discuss them.
  • Did authors check and confirm the amplification of AR with other materials such as biopsy for the study? If possible, they should confirm it in some cases.
  • If locations of AR-En and AR-E1, but not AR-E8, were amplified, amplified AR has a possibility to express AR variant protein including variant 7 which is reported to be associated with resistance to AR axis-targeted therapies. Did authors check expressions of AR variants in some patients?
  • Which material did authors define Glason score of 108 cases in mCRPC patients? With biopsy or total prostatectomy specimen?

Minor comment:

  • Authors should present the formal name of “OPHN1”.
  • In Table 1, “PSA atdate” should be changed to “PSA at date”.

Reviewer 2 Report

The paper describes a multiplex PCR assay to determine amplification of different regions of the androgen receptor (AR) in cell-free DNA isolated from mCRPC patients. The regions analyzed were: AR enhancer, exon 1, exon 8 or OPHN1 which is downstream of AR. A combination of 3 references controls was used for normalization. Amplification of the AR region was found in 41 out of 108 patients, in 9 cases all regions analyzed were amplified, in 6 cases only the enhancer region and in 4 cases only exon 8 was amplified. Survival association with amplification of the AR enhancer and exon 8 regions was shown. A multivariate Cox regression model showed  that AR exon 8 amplification, shorter time to androgen deprivation failure and higher serum alkaline phosphatase were independently associated with poor overall survival.

The data are solid and interesting and their main value is the establishment of a low-cost multiplex PCR assay to detect changes in AR gene loci levels in cell-free DNA from mCRPC patients. However, the plasma AR levels in CRPC patients have already been determined by many groups, see ref. 2-8 in the manuscript. The Kohli group also has already published on the prognostic value of AR amplification detected in cell-free DNA. Recently, digital PCR assays to analyze AR in cell-free DNA from patients were published by Sumiyoshi et al, Sci Reports, 2019, 9:4030 (not in the ref. list). This paper also shows that AR aberrations emerge with therapy resistance. 

An association between amplification of AR gene regions and overall survival is not new. There are less data on AR enhancer amplification. An interesting aspect of the results is that the different AR regions analyzed (enhancer, exon 1 and exon 8) do not behave exactly the same and that their survival association varies, but this is not really explored further or discussed. Also it would have been interesting to know whether there is a link to AR-V7 expression (which do not contain the exon 8 transcribed region).

Reviewer 3 Report

In the present paper authors evaluate prognostic value of multiplex digital PCR (dPCR) assay, detecting amplification of specific AR gene loci in metastatic castration-resistant prostate cancer (mCRPC).

The efforts of the authors are praiseworthy, the manuscript is well written, references are accurate. However, the manuscript shows several meaningful flaws that should be better addressed:

MAJOR ISSUES

Main limitation of this study is the presence of several missing clinical data that further reduce the prognostic assessment of study population and the results reliability. In particular, type of Androgen Deprivation Therapy and eventual second/third line treatments performed should be reported in “Clinical characteristics of patient samples” section. Moreover, it would have been of great clinical interest to explore Cancer Specific Survival (CSS) as well, since authors analyzed only Overall Survival, but no data are provided regarding baseline comorbidities or clinical features.

It would be useful integrating Table 1 with significant histological features such as primary Gleason score and ISUP grade at diagnosis. In this light, also number and location of bone metastases and presence of visceral metastases should be added.

Primary aim of the present study should be highlighted within the “abstract” section; in this light, considering “Patient samples” paragraph, inclusion criteria regarding study population should be better elucidated.  

MINOR ISSUES

As regards “Time from initiation of ADT to progression for CRPC stage” in Table 1, unit of measurement is missing.

Several typing errors are present along the manuscript. Page 1 line 39: “caner”. Please correct.

Author Response

Reviewer 3:

Major Comment 1:

“Main limitation of this study is the presence of several missing clinical data that further reduce the prognostic assessment of study population and the results reliability. In particular, type of Androgen Deprivation Therapy and eventual second/third line treatments performed should be reported in “Clinical characteristics of patient samples” section. Moreover, it would have been of great clinical interest to explore Cancer Specific Survival (CSS) as well, since authors analyzed only Overall Survival, but no data are provided regarding baseline comorbidities or clinical features.”

Response 1:

We have added further detail describing the clinical cohort characteristics to Table 1. ADT was administered by giving LHRH analogues exclusively, which has been added to the text in “Patient Characteristics”. Table 1 updates includes information on:

-number of patients in this cohort who had undergone previous radical prostatectomy or radiation therapy as primary prostate cancer therapy for localized stage disease;
-number of patients diagnosed upfront with advanced stage;
-if Gleason score was clinical (on biopsy) or pathological after prostatectomy;
-time from initial diagnosis to progression to castrate resistant state and to initiating chemotherapy;
-number of patients undergoing salvage or adjuvant therapies after progression post primary prostate organ specific therapies;
- number of patients with secondary hormonal therapies after progression post ADT
-number of patients who underwent systemic chemotherapy after secondary hormonal maneuvers in the mCRPC state; 

Addendum to Response 1

We missed addressing Reviewer 3’s comment on exploring Cancer Specific Survival.

As such, the state of the disease is advanced and metastatic castration resistant prostate cancer (CRPC) with a typical median survival time in this state being 2 to 3 years in prospective phase III randomized clinical trials published in the current decade. It is well accepted that death is due to disease and not competing causes. The Prostate Cancer Working Group 3 (PCWG3) suggestions, which are well accepted by experts outline outcome measures for prognostication by repeatedly mentioning death and not cancer specific survival. (Trial Design and Objectives for Castration-Resistant Prostate Cancer: Updated Recommendations From the Prostate Cancer Clinical Trials Working Group 3. Scher et al; https://www.ncbi.nlm.nih.gov/pubmed/26903579)

We attempted to adhere to these PCWG3 criteria including the PCWG3 suggestion of adding a measure of metastatic disease burden. For this (number of) bone metastatic sites is added in Table 1, which may permit more reliable prognostication along with the additional clinical characteristics that have now been added to describe the CRPC cohort.

Comment 2:
“It would be useful integrating Table 1 with significant histological features such as primary Gleason score and ISUP grade at diagnosis. In this light, also number and location of bone metastases and presence of visceral metastases should be added.”

Response 2:

Table 1 has been updated as mentioned above with the requested information to further detail the mCRPC cohort. Since this cohort was enrolled in advanced metastatic castrate resistant state, and primary prostate cancer treatments had taken place years earlier, we have included all early stage patient population treatments/characteristics that are available from previous medical records.

Comment 3:
“Primary aim of the present study should be highlighted within the “abstract” section; in this light, considering “Patient samples” paragraph, inclusion criteria regarding study population should be better elucidated.”

Response 3:

Thank you for pointing this out. We have added the suggested sentence in the abstract as below:
“108 mCRPC patients were recruited to test the association of specific AR loci amplification and clinical outcome”.

MINOR ISSUES
Comment 1:
“As regards “Time from initiation of ADT to progression for CRPC stage” in Table 1, unit of measurement is missing.”

Response 1:

We have added the measurement unit “month” in Table 1

Comment 2:
“Several typing errors are present along the manuscript. Page 1 line 39: “caner”. Please correct.”

Response 2:

Thank you for pointing this out. We have made these corrections wherever applicable.

Round 2

Reviewer 2 Report

In the revised version the authors now state that amplification of all three AR loci matched in most, but not all patients, which reflects the data presented. Generally they have satisfactorily addressed the queries of the reviewers.

Minor point:

Line 323: aa should be a

Reviewer 3 Report

Herein the Authors revised the paper entitled "Multiplex digital PCR to detect amplifications of specific androgen receptor loci in cell-free DNA for prognosis of metastatic castration-resistant prostate cancer".
In the revised version of the manuscript the Authors clarified all issues previously addressed and meaningfully improved the quality of the paper, as well as the strength of its clinical message.
Overall a good paper, scientifically correct, with good interpretation of the results.